# Local Smoothness in Variance Reduced Optimization

**Daniel Vainsencher, Han Liu**
**Dept. of Operations Research & Financial Engineering**
**Princeton University**
**Princeton, NJ 08544**
{`daniel.vainsencher,han.liu`}`@princeton.edu`

**Tong Zhang**
**Dept. of Statistics**
**Rutgers University**
**Piscataway, NJ, 08854**
`tzhang@stat.rutgers.edu`

## Abstract

We propose a family of non-uniform sampling strategies to provably speed up a class of stochastic optimization algorithms with linear convergence including Stochastic Variance Reduced Gradient (SVRG) and Stochastic Dual Coordinate Ascent (SDCA). For a large family of penalized empirical risk minimization problems, our methods exploit data dependent local smoothness of the loss functions near the optimum, while maintaining convergence guarantees. Our bounds are the first to quantify the advantage gained from local smoothness which are significant for some problems significantly better. Empirically, we provide thorough numerical results to back up our theory. Additionally we present algorithms exploiting local smoothness in more aggressive ways, which perform even better in practice.

## 1 Introduction

We consider minimization of functions of form

$$P(w) = n^{-1} \sum_{i=1}^{n} \phi_i \left( x_i^\top w \right) + R(w)$$

where the convex $\phi_i$ corresponds to a loss of $w$ on some data $x_i$, $R$ is a convex regularizer and $P$ is $\mu$ strongly convex, so that $P(w') \geq P(w) + \langle w' - w, \nabla P(w) \rangle + \frac{\mu}{2} \|w' - w\|^2$. In addition, we assume each $\phi_i$ is smooth in general and near flat in some region; examples include SVM, regression with the absolute error or $\varepsilon$ insensitive loss, smooth approximations of those, and also logistic regression.

Stochastic optimization algorithms consider one loss $\phi_i$ at a time, chosen at random according to a distribution $p^t$ which may change over time. Recent algorithms combine $\phi_i$ with information about previously seen losses to accelerate the process, achieving linear convergence rate, including Stochastic Variance Reduced Gradient (SVRG) [2], Stochastic Averaged Gradient (SAG) [4], and Stochastic Dual Coordinate Ascent (SDCA) [6]. The expected number of iterations required by these algorithms is of form $O\left( (n + L/\mu) \log \left( \varepsilon^{-1} \right) \right)$ where $L$ is a Lipschitz constant of all loss gradients $\nabla \phi_i$, measuring their smoothness. Difficult problems, having a condition number $L/\mu$ much larger than $n$, are called ill conditioned, and have motivated the development of accelerated algorithms [5, 8, 3]. Some of these algorithms have been adapted to allow importance sampling where $p^t$ is non uniform; the effect on convergence bounds is to replace the uniform bound $L$ described above by $L_{avg}$, the average over $L_i$, loss specific Lipschitz bounds.

In practice, for an important class of problems, a large proportion of $\phi_i$ need to be sampled only very few times, and others indefinitely. As an example we take an instance of smooth SVM, with $\mu = n^{-1}$ and $L \approx 30$, solved via standard SDCA. In Figure 1 we observe the decay of an upper bound on the updates possible for different samples, where choosing a sample that is white produces no update. The large majority of the figure is white, indicating wasted effort. For 95% of losses, the algorithm captured all relevant information after just 3 visits. Since the non white zone is nearly constant over time, detecting and focusing on the few important losses should be possible. This

represents both a success of SDCA and a significant room for improvement, as focusing just half the effort on the active losses would increase effectiveness by a factor of 10.

Similar phenomena occur under the SVRG and SAG algorithms as well. But is the phenomenon specific to a single problem, or general? for what problems can we expect the set of useful losses to be small and near constant?

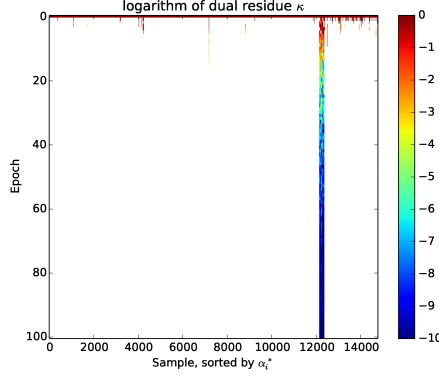

Figure 1: SDCA on smoothed SVM. Dual residuals upper bound the SDCA update size; white indicates zero hence wasted effort. The dual residuals quickly become sparse; the support is stable.

Allowing $p^t$ to change over time, the phenomenon described indeed can be exploited; Figure 2 shows significant speedups obtained by our variants of SVRG and SDCA. Comparisons on other datasets are given in Section 4. The mechanism by which speed up is obtained is specific to each algorithm, but the underlying phenomenon we exploit is the same: many problems are much smoother locally than globally. First consider a single smoothed hinge loss $\phi_i$, as used in smoothed SVM with smoothing parameter $\gamma$. The non-smoothness of the hinge loss is spread in $\phi_i$ over an interval of length $\gamma$, as illustrated in Figure 3 and given by

$$\phi_i(a) = \begin{cases} 0 & a > 1 \\ 1 - a - \gamma/2 & a < 1 - \gamma \\ (a-1)^2 / (2\gamma) & otherwise \end{cases}.$$

The Lipschitz constant of $\frac{d}{da}\phi_i(a)$ is $\gamma^{-1}$, hence it enters into the global estimate of condition number $L_{avg}$ as $L_i = \|x_i\| / \gamma$; hence approximating the hinge loss more precisely, with a smaller $\gamma$, makes the problems strictly more ill conditioned. But outside that interval of length $\gamma$, $\phi_i$ can be locally approximated as affine, having a constant gradient; into a correct expression of local conditioning, say on interval $B$ in the figure, it should contribute nothing. So smaller $\gamma$ can sometimes make the problem (locally) better conditioned. A set $I$ of losses having constant gradients over a subset of the hypothesis space can be summarized for purposes of optimization by a single affine

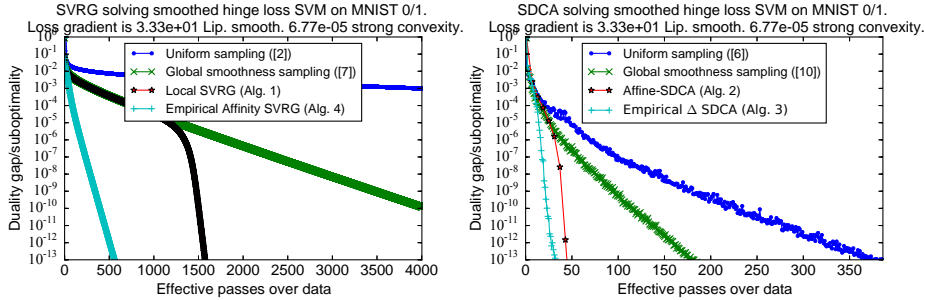

Figure 2: On the left we see variants of SVRG with $\eta = 1/(8L)$, on the right variants of SDCA.

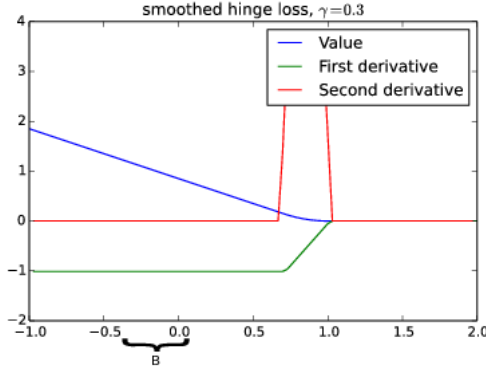

Figure 3: A loss $\phi_i$ that is near flat (Hessian vanishes, near constant gradient) on a "ball" $B \subset \mathbb{R}$. $B$ with radius $2r \, \|x_i\|$ is induced by the (Euclidean) ball of hypotheses $B(w^t, r)$, that we prove includes $w^*$. Then the loss $\phi_i$ does not contribute to curvature in the region of interest, and an affine model of the sum of such $\phi_i$ on $B$ can replace sampling from them. We find $r$ in algorithms by combining strong convexity with quantities such as duality gap or gradient norm.

function, so sampling from $I$ should not be necessary. It so happens that SAG, SVRG and SDCA naturally do such modeling, hence need only light modifications to realize significant gains. We provide the details for SVRG in Section 2 (the SAG case is similar) and for SDCA in Section 3.

Other losses, while nowhere affine, are locally smooth: the logistic regression loss has gradients with local Lipschitz constants that decay exponentially with distance from a hyperplane dependent on $x_i$. For such losses we cannot forgo sampling any $\phi_i$ permanently, but we can still obtain bounds benefitting from local smoothness for an SVRG variant.

Next we define formally the relevant geometric properties of the optimization problem and relate them to provable convergence improvements over existing generic bounds; we give detailed bounds in the sequel. Throughout $B(c, r)$ is a Euclidean ball of radius $r$ around $c$.

**Definition 1.** We shall denote $L_{i,r} = \max_{w \in B(w^*,r)} \left\| \nabla^2 \phi_i \left( x_i^\top \cdot \right) \right\|_2$ which is also the uniform Lipschitz coefficient of $\nabla \phi_i$ that hold at distance at most $r$ from $w^*$.

*Remark* 2. Algorithms will use similar quantities *not dependent on knowing* $w^*$ such as $\tilde{L}_{i,r}$ around a known $\tilde{w}$.

**Definition 3.** We define the average ball smoothness function $\mathcal{S} : \mathbb{R} \to \mathbb{R}$ of a problem by:

$$\mathcal{S}(r) = \sum_{i=1}^{n} L_{i,\infty} / \sum_{i=1}^{n} L_{i,r}.$$

In Theorem 5 we see that Algorithm 1 requires fewer stochastic gradient samples to reduce loss suboptimality by a constant factor than SVRG with importance sampling according to global smoothness. Once it has certified that the optimum $w^*$ is within $r$ of the current iterate $w^0$ it uses $\mathcal{S}(2r)$ times less stochastic gradient steps. The next measure similarly increases when many losses are affine on a ball around the optimum.

**Definition 4.** We define the ball affinity function $\mathcal{S} : \mathbb{R} \to [0, n]$ of a problem by:

$$\mathcal{A}(r) = \left( n^{-1} \sum_{i=1}^{n} 1_{\{L_{i,r} > 0\}} \right)^{-1}.$$

In Theorem 10 we see similarly that Algorithm 2 requires fewer accesses of $\phi_i$ to reduce the duality gap to any $\varepsilon > 0$ than SDCA with importance sampling according to global smoothness. Once it has certified that the optimum is within distance $r$ of the current primal iterate $w = w(\alpha^0)$ it accesses $\mathcal{A}(2r)$ times fewer $\phi_i$.

In both cases, local smoothness and affinity enable us to focus a constant portion of sampling effort on the fewer losses still challenging near the optimum; when these are few, the ratios (and hence

algorithmic advantage) are large. We obtain these provable speedups over already fast algorithms by using that local smoothness which we can certify. For non smooth losses such as SVM and and absolute loss regression, we can similarly ignore irrelevant losses, leading to significant practical improvements; the current theory for such losses is insufficient to quantify the speed ups as we do for smooth losses.

We obtain algorithms that are simpler and sometimes much faster by using the more qualitative observation that as iterates tend to an optimum, the set of relevant losses is generally stable and shrinking. Then algorithms can estimate the set of relevant losses directly from quantities observed in performing stochastic iterations, sidestepping the looseness of estimating $r$. There are two previous works in this general direction. The first paper work combining non-uniform sampling and empirical estimation of loss smoothness is [4]. They note excellent empirical performance on a variant of SAG, but without theory ensuring convergence. We provide similarly fast (and bound free) variants of SDCA (Section 3.2) and SVRG (Section 2.2). A dynamic importance sampling variant of SDCA was reported in [1] without relation to local smoothness; we discuss the connection in Section 3.

## 2 Local smoothness and gradient descent algorithms

In this section we describe how SVRG, in contrast to the classical stochastic gradient descent (SGD), naturally exposes local smoothness in losses. Then we present two variants of SVRG that realize these gains. We begin by considering a single loss when close to the optimum and for simplicity assume $R \equiv 0$. Assume a small ball $B = B(w, r)$ around our current estimate $w$ includes around the optimum $w^*$, and $B$ is contained in a flat region of $\phi_i$, and this holds for a large proportion of the $n$ losses.

SGD and its descendent SVRG (with importance sampling) use updates of form $w^{t+1} = w^t - \eta v_i^t / (p_i n)$, where $\mathbb{E}_{i \sim p} v_i^t / (p_i n) = \nabla F(w_t)$ is an unbiased estimator of the full gradient of the loss term $F(w) = n^{-1} \sum_{i=1}^{n} \phi_i(x_i^\top w)$. SVRG uses

$$v_i^t = \left( \nabla \phi_i \left( x_i^\top w^t \right) - \nabla \phi_i \left( x_i^\top \tilde{w} \right) \right) / (p_i n) + \nabla F(\tilde{w})$$

where $\tilde{w}$ is some reference point, with the advantage that $v_i^t$ has variance that vanishes as $w^t, \tilde{w} \to w^*$. We point out in addition that when $\tilde{w}, w^t \in B$ and $\nabla \phi_i \left( x_i^\top \cdot \right)$ is constant on $B$ the effects of sampling $\phi_i$ cancels out and $v_i^t = \nabla F(\tilde{w})$. In particular, we can set $p_i^t = 0$ with no loss of information. More generally when $\nabla \phi_i \left( x_i^\top \cdot \right)$ is near constant on $B$ (small $L_{i,r}$) the difference between the sampled values of $\nabla \phi_i$ in $v_i^t$ is very small and $p_i^t$ can be similarly small. We formalize this in the next section, where we localize existing theory that applied importance sampling to adapt SVRG statically to losses with varied global smoothness.

### 2.1 The Local SVRG algorithm

Halving the suboptimality of a solution using SVRG has two parts: computing an exact gradient at a reference point, and performing many stochastic gradient descent steps. The sampling distribution, step size and number of iterations in the latter are determined by smoothness of the losses. Algorithm 1, Local-SVRG, replaces the global bounds on gradient change $L_i$ with local ones $L_{i,r}$, made valid by restricting iterations to a small ball certified to contain the optimum. This allows us to leverage previous algorithms and analysis, maintaining previous guarantees and improving on them when $\mathcal{S}(r)$ is large.

For this section we assume $P = F$; as in the initial version of SVRG [2], we may incorporate a smooth regularizer (though in a different way, explained later). This allows us to apply the existing Prox-SVRG algorithm [7] and its theory; instead of using the proximal operator for fixed regularization, we use it to localize (by projections) the stochastic descent to a ball $B$ around the reference point $\tilde{w}$ see Algorithm 1. Then the theory developed around importance sampling and global smoothness applies to sharper local smoothness estimates that hold on $B$ (ignoring $\phi_i$ which are affine on $B$ is a special case). This allows for fewer stochastic iterations and using a larger stepsize, obtaining speedups that are problem dependent but often large in late stages; see Figure 2. This is formalized in the following theorem.

**Algorithm 1** Local SVRG is an application of ProxSVRG with $\tilde{w}$ dependent regularization. This portion reduces suboptimality by a constant factor, apply iteratively to minimize loss.

1. Compute $\tilde{v} = \nabla F(\tilde{w})$

2. Define $r = \frac{2}{\mu} \|\tilde{v}\|$, $R(w) = i_{B(\tilde{w},r)} = \begin{cases} 0 & w \in B(\tilde{w},r) \\ \infty & otherwise \end{cases}$ (by $\mu$ strong convexity, $w^* \in B(\tilde{w},r)$)

3. For each $i$, compute $\tilde{L}_{i,r} = \max_{w \in B(\tilde{w},r)} \nabla^2 \phi_i\left(x_i^\top w\right)$

4. Define a probability distribution: $p_i \propto \tilde{L}_{i,r}$, weighted Lipschitz constant $\tilde{L}_p = \max_i \tilde{L}_{i,r}/(np_i)$ and step size $\eta = \frac{1}{16\tilde{L}_p}$.

5. Apply the inner loop of Prox-SVRG:

   (a) Set $w^0 = \tilde{w}$

   (b) For $t \in \{1, \dots, m\}$:

      i. Choose $i^t \sim p$

      ii. Compute $v^t = \left(\nabla \phi_{i^t}\left(w^{t-1}\right) - \nabla \phi_{i^t}(\tilde{x})\right)/(np_{i^t}) + \tilde{v}$

      iii. $w^t = \text{prox}_{\eta R}\left(w^{t-1} - \eta v^t\right)$

   (c) Return $\hat{w} = m^{-1}\sum_{t \in [m]} w^t$

**Theorem 5.** *Let $\tilde{w}$ be an initial solution such that $\nabla F(\tilde{w})$ certifies that $w^* \in B = B(\tilde{w},r)$. Algorithm 1 finds $\hat{w}$ with*

$$\mathbb{E}F(\hat{w}) - F(w^*) \le (F(\tilde{w}) - F(w^*))/2$$

*using $O(d(n+m))$ time, where $m = \frac{128}{\mu}n^{-1}\sum_{i=1}^n L_{i,2r} + 3$.*

*Remark* 6. In the difficult case that is ill conditioned even locally so that $128n^{-1}\sum_{i=1}^n L_{i,2r} \gg n\mu$, the term $n$ is negligible and the ratio between complexities of Algorithm 1 and an SVRG using global smoothness approaches $\mathcal{S}(2r)$.

*Proof.* In the initial pass on the data, compute $\nabla F(\tilde{w})$, $r$ and $\tilde{L}_{i,r} \le L_{i,2r}$. We then apply a single round of Algorithm Prox-SVRG of [7], with the regularizer $R(x) = \chi_{B(\tilde{w},r)}$ localizing around the reference point. Then we may apply Theorem 1 of [7] with local $\tilde{L}_{i,r}$ instead of the global $L_i$ required there for general proximal operators. This allows us to use the corresponding larger stepsize $\eta = \frac{1}{16\tilde{L}_p} = \frac{1}{16n^{-1}\sum_{i=1}^n \tilde{L}_{i,r}}$. $\square$

*Remark* 7. The use of projections (hence the restriction to smooth regularization) is necessary because the local smoothness is restricted to $B$, and venturing outside $B$ with a large step size may compromise convergence entirely. While excursions outside $B$ are difficult to control in theory, in practice skipping the projection entirely does not seem to hurt convergence. Informally, stepping far from $B$ requires moving consistently against $\nabla F$, which is an unlikely event.

*Remark* 8. The theory requires $m$ stochastic steps per exact gradient to guarantee any improvement at all, but for ill conditioned problems this is often very pessimistic. In practice, the first $O(n)$ stochastic steps after an exact gradient provide most of the benefit. In this heuristic scenario, the computational benefit of Theorem 5 is through the sampling distribution and the larger step size. Enlarging the step size without accompanying theory often gains a corresponding speed up to a certain precision but the risk of non convergence materializes frequently.

While [2] incorporated a smooth $R$ by adding it to every loss function, this could reduce the smoothness (increase $\tilde{L}_{i,r}$) inherent in the losses hence reducing the benefits of our approach. We instead propose to add a single loss function defined as $nR$; that this is not of form $\phi_i\left(x_i^\top w\right)$ poses no real difficulty because Local-SVRG depends on losses only through their gradients and smoothness.

The main difficulty with the approach of this section is that in early stages $r$ is large, in part because $\mu$ is often very small ($\mu = n^{-\alpha}$ for $\alpha \in \{0.5, 1\}$ are common choices), leading to loose bounds

on $\tilde{L}_{i,r}$. In some cases the speed up is only obtained when the precision is already satisfactory; we consider a less conservative scheme in the next section.

## 2.2 The Empirical Affinity SVRG algorithm

Local-SVRG relies on local smoothness to certify that some $\Delta_i^t = \left\| \triangledown\phi_i \left( x_i^\top w^t \right) - \triangledown\phi_i \left( x_i^\top \tilde{w} \right) \right\|$ are small. In contrast, Empirical Affinity SVRG (Algorithm 4) takes $\Delta_i^t > t$ to be evidence that a loss is active; when $\Delta_i^t = 0$ several times, that is evidence of local affinity of the loss, hence it can be sampled less often. This strategy deemphasizes locally affine losses even when $r$ is too large to certify it, thereby focuses work on the relevant losses much earlier. Half of the time we sample proportional to the global bounds $L_i$ which keeps estimates of $\Delta_i^t$ current, and also bounds the variance when some $\Delta_i^t$ increases from zero to positive. A benefit of using $\Delta_i^t$ is that it is observed at every sample of $i$ without additional work. Pseudo code for the slightly long Algorithm 4 is in the supplementary material for space reasons.

## 3 Stochastic Dual Coordinate Ascent (SDCA)

The SDCA algorithm solves $P$ through the dual problem

$$D \left( \alpha \right) = -n^{-1} \sum_{i=1}^{n} \phi_i^* \left( -\alpha_i \right) + R^* \left( w \left( \alpha \right) \right)$$

where $w \left( \alpha \right) = \triangledown R^* \left( \frac{1}{\lambda n} \sum_{i=1}^{n} x_i \alpha_i \right)$. At each iteration, SDCA chooses $i$ at random according to $p^t$, and updates the $\alpha_i$ corresponding to the loss $\phi_i$ to increase $D$. This scheme has been used for particular losses before, and was analyzed in [6] obtaining linear rates for general smooth losses, uniform sampling and $l_2$ regularization, and recently generalized in [10] to other regularizers and general sampling distributions. In particular, [10] show improved bounds and performance by statically adapting to the global smoothness properties of losses; using a distribution $p_i \propto 1 + L_i \left( n\mu \right)^{-1}$, it suffices to perform $O \left( \left( n + \frac{L_{avg}}{\mu} \right) \log \left( \left( n + \frac{L_{avg}}{\mu} \right) \varepsilon^{-1} \right) \right)$ iterations to obtain an expected duality gap of at most $\varepsilon$. While SDCA is very different from gradient descent methods, it shares the property that when the current state of the algorithm (in the form of $\alpha_i$) already matches the derivative information for $\phi_i$, the update does not require $\phi_i$ and can be skipped. As we've seen in Figure 1, many losses converge $\alpha_i \to \alpha_i^*$ very quickly; we will show that local affinity is a sufficient condition.

### 3.1 The Affine-SDCA algorithm

The algorithmic approach for exploiting locally affine losses in SDCA is very different from that for gradient descent style algorithms; for some affine losses we certify early that some $\alpha_i$ are in their final form (see Lemma 9) and henceforth ignore them. This applies only to locally affine (not just smooth) losses, but unlike Local-SVRG, does not require modifying the algorithm for explicit localization. We use a reduction to obtain improved rates while reusing the theory of [9] for the remaining points. These results are stated for squared Euclidean regularization, but hold for strongly convex $R$ as in [10].

**Lemma 9.** *Let* $w^t = w \left( \alpha^t \right) \in B \left( w^*, r \right)$, *and let* $\{g_i\} = \bigcup_{w \in B(w^t, r)} \phi_i' \left( x_i^\top w \right)$; *in other words,* $\phi_i \left( x_i^\top \cdot \right)$ *is affine on* $B \left( w^t, r \right)$ *which includes* $w^*$. *Then we can compute the optimal value* $\alpha_i^* = -g_i$.

*Proof.* As stated in Section 7 of [6], for each $i$, we have $-\alpha_i^* = \phi_i' \left( x_i^\top w^* \right)$. Then if $\phi_i' \left( x_i^\top w \right)$ is a constant singleton on $B \left( w^t, r \right)$ containing $w^*$, then in particular that is $-\alpha_i^*$. □

The lemma enables Algorithm 2 to ignore a growing proportion of losses. The overall convergence this enables is given by the following.

**Algorithm 2** Affine-SDCA: adapting to locally affine $\phi_i$, with speedup approximately $\mathcal{A}(r)$.

1. $\alpha^0 = 0 \in \mathbb{R}^n$, $I^0 = \emptyset$.
2. For $\tau \in \{1, \dots\}$:
   (a) $\tilde{w}^\tau = w\left(\alpha^{(\tau-1)m}\right)$; Compute $r_\tau = \sqrt{2\left(P(\tilde{w}^\tau) - D\left(\alpha^{(\tau-1)m}\right)\right)/\mu}$
   (b) Compute $I^\tau = \left\{ i : \left| \bigcup_{w \in B(w^\tau, r)} \phi_i'\left(x_i^\top \tilde{w}^\tau\right) \right| = 1 \right\}$
   (c) For $i \in I^\tau \backslash I^{\tau-1}: \alpha_i^{(\tau-1)n} = -\phi_i'\left(x_i^\top \tilde{w}^\tau\right)$
   (d) $p_i^\tau \propto \begin{cases} 0 & i \in I^\tau \\ 1 + L_i(n\mu)^{-1} & otherwise \end{cases}$, $s_i = \begin{cases} 0 & i \in I^\tau \\ s/p_i^\tau & otherwise \end{cases}$
   (e) For $t \in [(\tau-1)m + 1, \tau m]$:
      i. Choose $i^t \sim p^\tau$
      ii. Compute $\Delta\alpha_{i^t}^t = s_{i^t} \cdot \left(\phi_{i^t}'\left(x_{i^t}^\top w(\alpha^t)\right) - \alpha_{i^t}^{t-1}\right)$
      iii. $\alpha_j^t = \begin{cases} \alpha_j^{t-1} + \Delta\alpha_j^t & j = i^t \\ \alpha_j^{t-1} & otherwise \end{cases}$

**Theorem 10.** *If at epoch $\tau$ Algorithm 2 is at duality gap $\varepsilon^\tau$, it will achieve expected duality gap $\varepsilon$ in at most $\left(n' + \frac{\mathcal{A}^{-1}(2r)L_{avg}'}{\mu}\right) \log\left(\left(n' + \frac{\mathcal{A}^{-1}(2r)L_{avg}'}{\mu}\right)\frac{\varepsilon^\tau}{\varepsilon}\right)$ iterations, where $n' = n - |I^\tau|$ and $L_{avg}' = \frac{n'^{-1}\sum_{i \in [n]\backslash I^\tau} L_i}{\mu}$.*

*Remark* 11. Assuming $L_i = L$ for simplicity, and recalling $\mathcal{A}(2r) \leq n/n'$, we find the number of iterations is reduced by a factor of at least $\mathcal{A}(2r)$, compared to using $p_i \propto 1 + L_i(n\mu)^{-1}$. In contrast, the cost of the steps 2a to 2d added by Algorithm 2 is at most a factor of $O((m+n)/m)$, which may be driven towards one by the choice of $m$.

Recent work [1] modified SDCA for dynamic importance sampling dependent on the so called dual residual:

$$\kappa_i = \alpha_i + \phi_i'\left(x_i^\top w(\alpha)\right)$$

(where by $\phi_i'(w)$ we refer to the derivative of $\phi_i$ at $w$) which is 0 at $\alpha^*$. They exhibit practical improvement in convergence, especially for smooth SVM, and theoretical speed ups when $\kappa$ is sparse (for an impractical version of the algorithm), but [1] does not tell us when this pre-condition holds, nor the magnitude of the expected benefit in terms of properties of the problem (as opposed to algorithm state such as $\kappa$). In the context of locally flat losses such as smooth SVM, we answer these questions through local smoothness: Lemma 9 shows $\kappa_i$ tends to zero for losses that are locally affine on a ball around the optimum, and the practical Algorithm 2 realizes the benefit when this certification comes into play, as quantified in terms of $\mathcal{A}(r)$.

### 3.2 The Empirical $\triangle$ SDCA algorithm

Algorithm 2 uses local affinity and a small duality gap to certify the optimality of some $\alpha_i$, avoiding calculating $\Delta\alpha_i$ that are zero or useless; naturally $r$ is small enough only late in the process. Algorithm 3 instead dedicates half of samples in proportion to the magnitude of recent $\Delta\alpha_i$ (the other half chosen uniformly). As Figure 2 illustrates, this approach leads to significant speed up much earlier than the approach based on duality gap certification of local affinity. While we it is not clear that we can prove for Algorithm 3 a bound that strictly improves on Algorithm 2, it is worth noting that except for (probably rare) updates to $i \in I^\tau$, and a factor of 2, the empirical algorithm should quickly detect all locally affine losses hence obtain at least the speed up of the certifying algorithm. In addition, it naturally adapts to the expected small updates of locally smooth losses. Note that $\Delta\alpha_i$ is closely related to (and might be replaceable by) $\kappa$, but the current algorithm differs significantly from those in [1] in how these quantities are used to guide sampling.

**Algorithm 3** Empirical $\Delta$ SDCA

1. $\alpha^0 = 0 \in \mathbb{R}^n$, $A_i^t = 0$.

2. For $\tau \in \{1, \dots\}$:

    (a) $p^\tau = 0.5p^{\tau,1} + 0.5p^2$ where $p_i^{\tau,1} \propto A_i^{(\tau-1)m}$ and $p_i^2 = n^{-1}$

    (b) For $t \in [(\tau-1)\,m+1, \tau m]$:

        i. Choose $i^t \sim p^\tau$
        ii. Compute $\Delta\alpha_{i^t}^t = s_{i^t} \cdot \left(\phi'_{i^t}\left(x_{i^t}^\top w\left(\alpha^t\right)\right) - \alpha_{i^t}^{t-1}\right)$
        iii. $A_j^t = \begin{cases} 0.5A_j^{t-1} + 0.5\left|\Delta\alpha_j^t\right| & j = i^t \\ A_j^{t-1} & otherwise \end{cases}$
        iv. $\alpha_j^t = \begin{cases} \alpha_j^{t-1} + \Delta\alpha_j^t & j = i^t \\ \alpha_j^{t-1} & otherwise \end{cases}$

## 4 Empirical evaluation

We applied the same algorithms with almost[1] the same parameters to 4 additional classification datasets to demonstrate the impact of our algorithm variants more widely. The results for SDCA are in Figure 4, those for SVRG in Figure 5 in Section 7 in the supplementary material for lack of space.

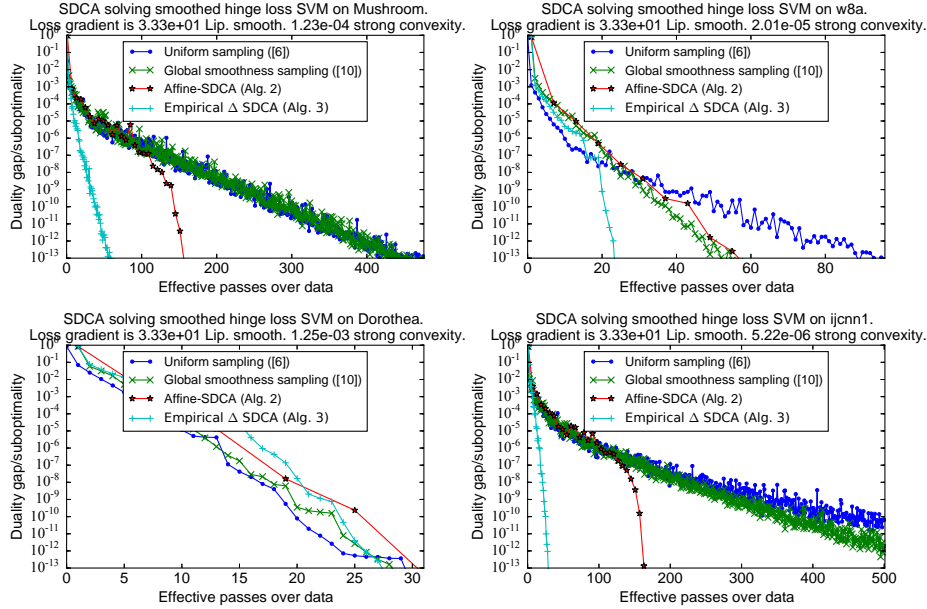

Figure 4: SDCA variant results on four additional datasets. The advantages of using local smoothness are significant on the harder datasets.

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
