[Supplementary Material]

# 5 Empirical Affinity SVRG

**Algorithm 4** Empirical Affinity SVRG algorithm. The parameters $c_1 > 1 > c_2$ were tuned for the MNIST dataset to $1.2, 0.8$, and reused elsewhere.

1. $w^0 = \hat{w}^0 = 0 \in \mathbb{R}^d$, $A_i^0 = 0 \; \forall i \in [n]$.
2. For $\tau \in \{1, \dots\}$:
   (a) $\tilde{w}^\tau = \hat{w}^\tau$
   (b) Compute $\tilde{v}^\tau = \nabla F(\tilde{w}^\tau)$
   (c) Define $r^\tau = \frac{2}{\mu} \|\tilde{v}^\tau\|$ (by $\mu$ strong convexity, $w^* \in B(\tilde{w}^\tau, r)$)
   (d) For each $i$, compute $\tilde{L}_{i,r}^\tau = \max_{w \in B(\tilde{w}^\tau, r)} \nabla^2 \phi_i(x_i^\top w)$
   (e) $p_i^\tau \propto \left(p_i^{\tau,1} + p_i^{\tau,2}\right)/2$, where $p_i^{\tau,1} \propto \tilde{L}_{i,r}^\tau$ and $p_i^{\tau,2} \propto A_i^{(\tau-1)m}$
   (f) $\hat{L}_i^\tau = \begin{cases} \tilde{L}_{i,r} & \tau = 1 \\ A_i^t & otherwise \end{cases}$
   (g) $\eta^\tau = 1/\left(8 \max_i \left(\hat{L}_i^\tau / (np_i^\tau)\right)\right)$
   (h) For $t \in [(\tau-1)m+1, \tau m]$:
      i. Choose $i^t \sim p^\tau$
      ii. $\Delta_{i^t}^t = \nabla \phi_{i^t}(w^{t-1}) - \nabla \phi_{i^t}(\tilde{x})$
      iii. $A_j^t = \begin{cases} c_1 \tilde{L}_{j,r}^\tau & j = i^t; \left|\Delta_j^t\right| > 0 \\ c_2 A_j^{t-1} & j = i^t; \Delta_j^t = 0 \quad \forall j \in [n] \\ A_j^{t-1} & otherwise \end{cases}$
      iv. $v^t = \Delta_{i^t}^t / (np_{i^t}) + \tilde{v}$
      v. $w^t = w^{t-1} - \eta v^t$
   (i) $\hat{w}^\tau = m^{-1} \sum_{t \in [(\tau-1)m+1, \tau m]} w^t$

# 6 Proof of Theorem 10

*Proof.* In step $2a$ we identify $B = B(\tilde{w}^\tau, r)$ such that $w^* = w(\alpha^*) \in B$. Then using Lemma 9, the set of points $I_\tau$ defined in step $2b$ have $\alpha_i^{(\tau-1)m}$ set to $\alpha_i^*$ in step $2c$, and that correct value is retained for all $t \geq (\tau-1)m$ because $p_i^\tau = 0$ henceforth. Also, $\phi_i$ are affine on $B$. Now consider a problem that is equivalent on $B$, where $\phi_i$ for $i \in I^\tau$ are replaced by their affine approximations at $\tilde{w}^\tau$:

$$\overline{\phi}_i(w) = \begin{cases} \phi_i(\tilde{w}^\tau) + (w - \tilde{w}^\tau)^\top \nabla \phi_i(\tilde{w}^\tau) & i \in I^\tau \\ \phi_i & otherwise \end{cases};$$

the conjugates $\overline{\phi}_i^*$ of the affine approximations admit (are finite on) only the correct value $\alpha_i^*$. Then the iterations of Algorithm 2 in substeps of $2e$ behave exactly like Iprox-SDCA of [9] (the full version of [10]) applied to the modified problem. Now note that since Iprox-SDCA assigns $p_i \propto 1 + L_i (n\mu)^{-1}$, and $\overline{\phi}_i$ have $L_i = 0$ on $I^\tau$, hence ignores them completely, progress of Iprox-SDCA on the modified problem is equal to that on a reduced modified problem from which $I^\tau$ are removed entirely (having $P'(w) = n^{-1} \sum_{i \in [n] \setminus I^\tau} \overline{\phi}_i(w) + R(w)$). By Lemma 2 of [9] and weak duality, we have

$$\mathbb{E}D'(\alpha^*) - D'(\alpha^t) \geq \mathbb{E}D'(\alpha^{t+1}) - D'(\alpha^t) \geq \frac{s}{n'} \left[P'(w(\alpha^t)) - D'(\alpha^t)\right] \geq \frac{s}{n'} \left[D'(\alpha^*) - D(\alpha^t)\right].$$

Then on one hand it is enough to achieve $\varepsilon \geq \frac{n'}{s} [D'(\alpha^*) - D'(\alpha^t)] \geq P'(w(\alpha^t)) - D(\alpha^t)$, and on the other $D'(\alpha^*) - \mathbb{E}D'(\alpha^{t+1}) \leq \left(1 - \frac{s}{n'}\right)(D'(\alpha^*) - D'(\alpha^t))$ and hence $D'(\alpha^*) - \mathbb{E}D'(\alpha^{t+k}) \leq \left(1 - \frac{s}{n'}\right)^k (D'(\alpha^*) - D'(\alpha^t)) \leq \exp\left(-\frac{sk}{n'}\right)(D'(\alpha^*) - D'(\alpha^t))$. Then it is

enough to have $\frac{n'}{s} \exp\left(-\frac{sk}{n'}\right) \varepsilon^\tau \leq \varepsilon \iff k \geq \frac{n'}{s} \log\left(\frac{n'\varepsilon^\tau}{s\varepsilon}\right)$. Applying Proposition 2 of [9] to the reduced and modified problem, we obtain

$$\frac{n'}{s} = n' + \mu^{-1}\left(n^{-1} \sum_{i \in [n]\setminus I^\tau} L_i\right).$$

$\square$

## 7 Empirical results for SVRG

Figure 5: SVRG variant results on four additional datasets. For the Mushroom dataset, the global plot occludes the uniform.