[Reviews · NeurIPS 2015]

Submitted by Assigned_Reviewer_1

The authors propose a non-uniform sampling scheme for variance reduced SGD type algorithms based on local smoothness and the fact that the gradient of many individual losses is constant. The authors show that such a scheme is able to outperform uniform sampling for SVRG and SDCA.

Overall the idea is an interesting one and seems to perform well in practice. However, I feel that the paper has some major clarity issues.

In general, I find the paper quite difficult to read. Quantities and algorithms are not properly defined before they appear in the text. Eg in figure 2 the different procedures are not fully explained.

Importantly, the main results are stated in quite unfamiliar ways (i.e. the time taken to halve the error or to exponentially reduce the duality gap). As such they are difficult to compare to standard results of the base methods. A detailed discussion of this is missing from the paper.

The notion of distance in B is never properly described. I assume it is l2 distance but does it matter?

What if a different measure of distance is used?

Figure 3 is confusing. B is defined as some kind of distance in parameter space but here it is used with respect to the loss phi_i. Much of the confusion stems from the figure not being properly labeled or explained.

The paper is essentially missing any discussion or summary particularly of the empirical results. This is troublesome because I don't understand what "drop certified correct" means and why the proposed scheme performs worse on the Dorothea dataset.

Typos. 175: SDG = SGD? 374: rate = rare?
Summary: The overall idea is interesting and definitely worth following up. But currently, the paper could stand a lot more editing for clarity and indepth discussion about what the theoretical and empirical results mean.

Submitted by Assigned_Reviewer_2

Comments following author response:

Thank you for the description of how to construct the ball and estimate the Lipschitz constants, please add this to the paper.

(This was a requested 'light' review.)

This work considers accelerating the convergence rate of SAG/SVRG/SDCA by using local properties of the function around its optimum. This can lead to very large practical performance improvements. This is well-known in the context of batch-gradient methods, and was previously demonstrated in context of SAG in the experiments of [4].

In the case of SVRG, a ball around the optimum is defined and non-uniform sampling based on the Lipschitz constant over the ball is used to improve the convergence rate. The theory for the SVRG approach is trivial, with the only non-obvious part being that the iterates are projected onto the ball (which is assumed to be known). The novelty of this idea is limited a bit because a similar idea (adapting to the local Lipschitz constants) is also discussed in [4] in the context of SAG and is used in their code. This should be discussed in the paper and the experiments should compare to this strategy.

In the case of SDCA, the method avoids sampling functions which are affine over the ball. This is a very restrictive assumption, but it includes many variants of SVMs. For, the theory requires a bit of work but is still very simple and largely relies on the results in [9].

The authors propose variants of these ideas that do not follow from the theory but that work better in practice, but again these should be compared to prior approaches that propose similar ideas.

The paper also needs to discuss how the ball and the local Lipschitz constants can be computed in practice. It seems that this is only possible in certain restricted settings. The paper seems incomplete without this discussion.
Summary: The work addresses an important practical problem, and proposes a strategy that can increase practical performance by a substantial amount.

However, the novelty of taking advantage of local smoothness is somewhat over-stated (without a comparison to prior methods that do similar things) and the theory is fairly straightforward. The work is also missing a discussion of how to practically implement the solution ball and local Lipschitz constants in practice.

Submitted by Assigned_Reviewer_3

All variance reducing stochasting algorithms use a sampling distribution over the training points. Though the first works used a uniform distribution, it has quickly been extended to distributions based on the Lipschitz constants of the gradients of each individual function. These constants, however, were computed once for the entirety of the optimization.

Orthogonally, the optimal stepsize for these methods relies on the maximum of these constants. Schmidt et al proposed a line-search for the SAG algorithm which adapted this stepsize throughout the optimization.

The current paper merges these two techniques in some way by adapting the sampling distribution along the optimization. Though obvious in retrospect, I do not know of any work implementing this idea and making it work.

I think Algorithms 1 and 2 are of very limited interest since they apply in very contrived cases. In particular, I think the authors should make clearer than, in most cases, these algorithms will not bring anything over the classical ones (a brief mention is made at the bottom of page 5).

I am however impressed by the results obtained by empirical SVRG and this enough makes it worth publicizing. It is unfortunate that these results end up in the appendix.

Overall, my main comment would be to rewrite this paper as a practical implementation with promising results and expand the experimental section, while shrinking the more theoretical part which is obvious.

Other comments: - The beginning of section 2 with the SGD (typo) and SVRG updates is quite confusing. Please reword it. - I did not see mentioned the cost of drawing from a non-uniform distribution. It seems to be that, to be efficient, you need to build a tree of the probabilities. Do you need to rebuild the tree everytime a local L changes?

In conclusion, I'm leaning for acceptance because of the impressive results but I think it should definitely be recast as an empirical paper.
Summary: The practical results are impressive, the theoretical ones are not. Since the issues of varying smoothness is of practical relevance in optimization, I'm leaning towards acceptance provided the paper is recast as an empirical paper.

Submitted by Assigned_Reviewer_4

This paper introduces methods for improving convergence of stochastic variance reduced gradient (SVRG) and stochastic dual coordinate ascent (SDCA) methods. This is based on the observation that the individual loss functions are smoother locally than globally.

In the common optimization problems in machine learning,

the cost function can be written as a sum of several loss functions. It is an interesting idea to adapt the sampling procedure in stochastic methods based on local smoothness property.

It is very hard to read the paper and follow the arguments presented by the authors. The authors refer to the quantities and concepts that they have never introduced. Just to mention a few instances: 1) In the first paragraph of introduction, the authors refer to the $\mu$-strongly convex functions. This concept is a key definition and it should be defined in the paper. 2) At line 80, the authors talk about smoothed SVM with smoothing parameter $\gamma$ without mentioning what this parameter mean. I suggest that the authors rewrite the manuscript to make it self-contained and easily readable.

In the simulation results, the authors show the improvement obtained by their proposed approaches. This improvement is significant only in the later stages of effective passes through the data. Stochastic methods make the most progress in the initial steps when the largest reduction in the test error is observed. It is, therefore, important that the authors also show the performance on a test set.
Summary: Although the ideas explained in the paper are interesting, the current status of exposition of those ideas and corresponding simulation results are not sufficient for a nips submission.

Author Feedback
Author rebuttal: We thank the reviewers for helping improve our paper. Most issues of clarity had easy fixes (some addressed towards the end), we focus on other issues here.

Reviewer 4, you pointed out that our reference [4] actually did experiments taking advantage of local smoothness in a stochastic setting. Thank you very much! we had missed them, restricted as they are to the sections "Implementation Details" (4.6 and 4.8) and "Experimental Results" (5.5). The algorithm variants described therein use techniques similar to those of our Algorithms 3 and 4: they estimate local smoothness rather than bounding it. These variants are not accompanied by theoretical results, despite the fact that increasing step size can prevent convergence.

Most of you have noted that the theoretical results we provide are not surprising or novel in technique, and this is true. Nevertheless guaranteeing that convergence is preserved and proving a strict advantage for appropriate problems can help move exploitation of local smoothness from "helpful on at least a few datasets" to being a proven part of the toolbox of stochastic optimization.

Further contributions not found in [4] are:
- Exploitation of local smoothness ideas on multiple distinct algorithms (SDCA and SVRG),
- Identification of a common idea to these algorithms beyond variance reduction (a joint linear model of all flat losses is cheap and accurate; l.85-l.88).
- Empirical evidence of 90% wasted effort in standard algorithms on SVM-like problems specifically due to iterating on locally flat losses (Fig. 1), and hence
- Helping identify when exploiting local smoothness will make a difference.

Reviewers 1 and 4: First, the ball B is in every algorithm chosen around a point known by the algorithm (reference point in SVRG, the current primal solution in SDCA). The radius is chosen so that the optimum is always inside B, but this does not assume we know the optimum; it follows from strong convexity (see for example line 220 in Algorithm 1). The ball B uses the Euclidean distance induced by the inner product applied to w and the data x_i. This particular distance is useful, because then restricting w to B restricts w^\top x_i to a segment of length 2*r*||x_i|| of the corresponding loss. ||x_i|| times the maximum Lipschitz of the derivative of the loss on that segment is exactly the L_{i,r} we need. We now explain this in detail in the supplementary material, in addition to the illustration of this in line 80 for the smooth hinge loss and in Figure 3. Reviewer 1: Figure 3 confused the parameter of the loss with the space of w; this inconsistency is now fixed by taking for the sake of the example x_i = 1.

Reviewer 2, you suggested to improve the clarity of first part of Section 2 which gives an intuitive explanation for how SVRG and SAG can exploit local smoothness. We changed the ball $B$ which in this here was around the optimum to merely include the optimum as in the algorithms, and reduced the mention of variance reduction to focus on the effect of local smoothness. If something else was not clear, specifics would help us.

About updating sampling distributions: our algorithms recompute those only every few passes over the data. For algorithms 3 and 4 which do not read x_i beyond necessary for the updates, the costs are negligible, and updating the sampling probability and steps size more continuously could improve performance, but we have not done this.

We have of course corrected the typos, identified each method in the experiments with a reference, and defined terms mentioned by Reviewer 3.